# Extinction debts and colonization credits of non-forest plants in the European Alps

Sabine B. Rumpf [1], Karl Hülber [1,2], Johannes Wessely[1], Wolfgang Willner [1,2], Dietmar Moser[1], Andreas Gattringer [1], Günther Klonner[1], Niklaus E. Zimmermann [3,4] & Stefan Dullinger [1]

Mountain plant species shift their elevational ranges in response to climate change. However, to what degree these shifts lag behind current climate change, and to what extent delayed extinctions and colonizations contribute to these shifts, are under debate. Here, we calculate extinction debt and colonization credit of 135 species from the European Alps by comparing species distribution models with 1576 re-surveyed plots. We find extinction debt in 60% and colonization credit in 38% of the species, and at least one of the two in 93%. This suggests that the realized niche of very few of the 135 species fully tracks climate change. As expected, extinction debts occur below and colonization credits occur above the optimum elevation of species. Colonization credits are more frequent in warmth-demanding species from lower elevations with lower dispersal capability, and extinction debts are more frequent in cold-adapted species from the highest elevations. Local extinctions hence appear to be already pending for those species which have the least opportunity to escape climate warming.

[1] Department of Botany and Biodiversity Research, University of Vienna, Rennweg 14, 1030 Vienna, Austria. [2] Vienna Institute for Nature Conservation and Analyses, Gießergasse 6, 1090 Vienna, Austria. [3] Swiss Federal Research Institute WSL, Zürcherstrasse 111, 8903 Birmensdorf, Switzerland. [4] Department of Environmental Systems Science, Swiss Federal Institute of Technology ETH, Universitätstrasse 16, 8006 Zürich, Switzerland. Correspondence and requests for materials should be addressed to S.B.R. (email: sabine.rumpf@univie.ac.at)

Many mountain biota have shifted their distribution upslope during the recent decades, and these shifts are hypothesized to be at least partly driven by climate warming[1–4]. So far, the most prominent footprint of plant species' range dynamics is an accelerating increase of species richness on mountain summits[5,6]. While many species previously growing at lower elevations have been colonizing mountain tops, few of the species initially present have gone locally extinct so far[7]. This development seems to contradict the results of species distribution models (SDMs) which foresee massive range loss and widespread local extinction of plant species under a warming climate, especially at highest elevations[8,9]. Observations and models may disagree because the latter miss important variables and processes such as biotic interactions[10] or species plasticity[11,12]. In addition, observations represent snapshots of transient dynamics, while models predict steady states between the distribution of species and future climatic conditions and cannot usually inform whether or when such a new equilibrium may be reached[13]. The apparent contradiction between data and models may hence result from lags in the response of populations to the ongoing change in climatic conditions[14,15]. Based on such lags, species distributions that are in disequilibrium with current climatic conditions are indeed expected for the next decades and even centuries[16]. However, whether such disequilibria have already emerged in mountain plant distributions during recent decades is under debate[2,3,5,14,17].

Range shifts under environmental change are driven by two major processes: the colonization of sites that become newly suitable and the extinction of existing populations at sites that become unsuitable. Both of these processes may lag behind, that is, be in disequilibrium with the changing environment, with delayed colonization resulting in an immigration or colonization credit, and delayed extinction in an extinction debt, respectively[18,19]. Various peculiarities of alpine plants and environments such as low rates of germination and seedling establishment, pronounced clonality and longevity, slow rate of soil formation, considerable dispersal limitation by natural habitat fragmentation, and high impact of biotic interactions on species distributions suggest that both colonization credits and extinction debts could be widespread in mountain vegetation under climate change[14]. Whether delay times are more likely and/or longer for colonization or extinction processes is, however, unknown. This is unfortunate because estimating credits and debts may allow for more realistic predictions of the eventual fate of species and communities under climate change[20], and may have important ramifications for conservation[18,21].

In mountain ecosystems, colonization events are intuitively expected at upper and extinctions at lower range limits when climate warms[2,22,23]. As a consequence, credits and debts should occur at these opposite range limits, in a similar way to what has been shown along latitudinal gradients[20]. However, it has also been hypothesized that due to the high variability in microclimatic conditions in alpine environments, new habitat might emerge close-by under climate change[24,25]. As a corollary, both colonization credits and extinction debts could occur across the entire range of species without a clear stratification of these phenomena along the elevational gradient.

Here, we use a combination of re-survey data and species distribution modeling to detect possible disequilibrium dynamics in the non-forest vegetation of the European Alps during recent decades. Instead of assessing whether the species composition of local plant assemblages kept track with the changing climate[17,26], we analyze whether the distribution of individual species has changed as expected from climatic trends, because emerging disequilibria are most likely species-specific[27]. We therefore do not compare differences between the upslope shifts observed and those expected from adiabatic lapse rates[2,3] (that is, from the linear decrease of temperature with elevation), because such an approach disregards other potentially important factors such as precipitation changes[28], and does not account for the topography of the terrain[24]. Instead, we fit SDMs of species based on several environmental variables and then use these SDMs to project patterns of expected recent distribution changes across a set of 1576 semi-permanent vegetation plots that have been sampled historically (before 1970) and re-sampled in 2014 and 2015. We quantify the number of expected colonization and extinction events as the number of plots where these processes are predicted to occur by the SDMs, and the number of colonization credit and extinction debt events as the number of plots where these processes are predicted to occur by the SDMs but have not actually been observed in the re-survey. More specifically, we analyze whether the current distribution of Alpine species in the study area matches the distribution expected from their historical climatic niches. Our results suggest that the realized niche of few species has fully tracked climate change. Thirty-eight percent of the species have not yet realized all predicted colonizations and 60% have not gone extinct from all sites that have become unsuitable to them. As expected under a general upslope shift under climate change[1–4], there is an elevational structure in the frequency of these unrealized events. Extinction debts are predominating towards the lower, and colonization credits towards the upper range limit of mountain plants. The amount of these extinction debts and colonization credits can be related to species-specific properties. The emergence of colonization credits depends on species' dispersal capabilities[14,16,29] but, contrary to our expectations[14,16,19], extinction debts are not related to species' persistence capabilities. Yet, extinction debts are more frequent in cold-adapted species from the highest elevations, and colonization credits are more frequent in warmth-demanding species from lower elevations. Local extinctions hence appear to be already pending for those species which have the least opportunity to escape climate warming. These results therefore do not relax concerns about potentially detrimental effects of climate change[30] on alpine biodiversity and suggest that even if climate change were halted, its effects would be likely to continue.

## Results

**Frequency of extinction debts and colonization credits.** Based on the predictions of our models, extinction and colonization events were expected for all 135 plant species in our dataset, and only nine and four species had expected colonization and extinction events in less than 10 plots, respectively (Supplementary Fig. 1). We detected either extinction debt or colonization credit events in 93% of the plant species studied. Sixty six percent of the species exhibited an extinction debt and 38% had a colonization credit. However, if species' had extinction debts, they were, on average, less frequent (16% of expected extinction events per species were not realized) than colonization credits (54% of the expected colonization events were not realized). On average, the 135 species had a mean extinction debt of 10% and a colonization credit of 20%, i.e., 10 and 20% of the expected extinction and colonization events did not occur, respectively, but owing to considerable species-specific variation in the magnitude of these lags, this difference was not significant (Wilcoxon signed-rank test, $p$-value = 0.088; Fig. 1). Moreover, extinction debts and colonization credits were only weakly correlated (Pearson's $r = 0.42$). Eighty-three percent of the plant species in our data set had either only extinction debts or only colonization credits, while only 10% had both types of disequilibrium (inset Fig. 1).

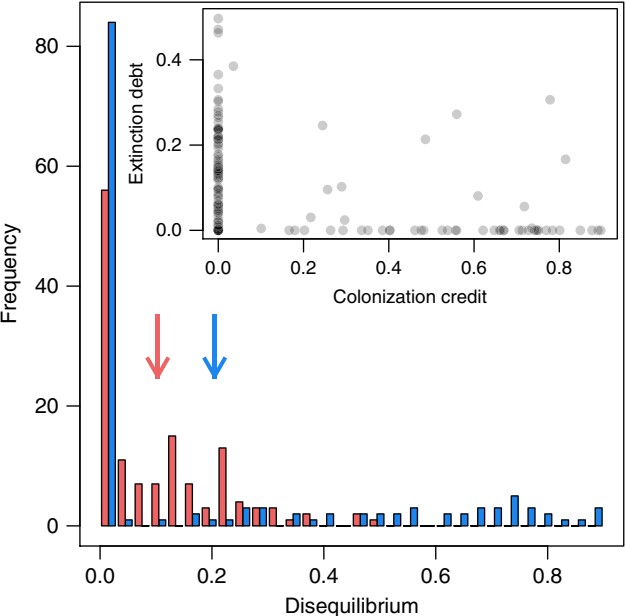

**Fig. 1** Frequency of extinction debts and colonization credits of Alpine plant species. Values of disequilibria are relative to expected extinction and colonization events, respectively (see Methods section). In the main figure, histograms are depicted for extinction debts (red) and colonization credits (blue) with mean values as vertical arrows ($n = 135$ species). The inset illustrates the correlation of extinction debts and colonization credits. Here, species are depicted as dots and are darker with more species at the same position. Source data are provided as a Source Data file

**Elevational distribution of ecological disequilibria.** On average, observed colonization events were located 80 m higher than the historical elevational optima of the respective species (paired *t*-test with Bonferroni correction; df = 132, $t = 7.57$, $p < 0.001$) and colonization credit events were located even 52 m higher upslope than observed colonization events (paired *t*-test with Bonferroni correction; df = 132, $t = 4.82$, $p < 0.001$). By contrast, both observed extinctions and extinction debts were situated below the species' historical elevational optima (paired *t*-test with Bonferroni correction; observed extinctions: 76 m, df = 134, $t = -7.52$, $p < 0.001$; extinction debts: 73 m, df = 127, $t = -4.98$, $p < 0.001$). However, extinction debt events did not differ in elevation from observed extinction events (paired *t*-test with Bonferroni correction; 5 m, df = 127, $t = 0.28$, $p = 1.000$; Fig. 2).

**Relations to species-specific properties.** Detected disequilibria were related to several of the species properties considered, although the variance explained was only moderate in some, and low in all other cases (Supplementary Table 1). We found a significant negative relation between species' dispersal capability and the amount of occurring colonization credits, i.e., species able to disperse seeds over greater distances had lower colonization credits (Fig. 3a, Supplementary Table 1). On the contrary, we could detect no relation between dispersal capability and extinction debts (Fig. 3a, Supplementary Table 1), nor between persistence capability and either type of disequilibrium (Fig. 3b, Supplementary Table 1). Furthermore, cryophilic species and species from higher elevations had higher levels of extinction debts (Fig. 3c, d, Supplementary Table 1). By contrast, thermophilic species and species from lower elevations had higher colonization credits (Fig. 3c, d, Supplementary Table 1).

## Discussion

While modeling studies have predicted considerable lags in the future response of Alpine plants to climate warming[15], empirical data about dynamics of the recent past appear contradictory. Some studies concluded that mountain vegetation follows climate almost without delay[5,17], whereas other results indicated considerable inertia, especially in case of subalpine and alpine grassland vegetation[31,32]. Importantly, most empirical analyses available so far took a 'community perspective', i.e., they analyzed how local assemblages of plants have reacted to climate change in terms of attributes like species richness or composition[5,17]. The few studies that focused on the dynamics of individual species commonly found important lags, but these studies have been restricted to comparing observed shifts in elevational meters to those expected from adiabatic lapse rates[2,3,22]. The results presented here corroborate these latter findings qualitatively in that they demonstrate that disequilibria are currently widespread among mountain plants. However, our combination of re-survey data with species distribution modeling allows for a more detailed assessment of these disequilibria.

Sixty percent of species have not gone extinct from all sites that our models classified as no longer suitable at the time of the re-survey, 38% have not been able to colonize all the sites that, according to our models, have become suitable to them, and 93% showed at least one type of lagged response. These results suggest that rapid response of community attributes like species richness[5,6] probably masks considerable lag times in the reaction of individual species to climate warming, even if the temporal magnitude of these lags is not deducible from our results. Although considerable, the frequency of these debts may even appear low compared to the pronounced lags that other observational studies detected with respect to changes in elevational species ranges[2,3] and the low extinction rates reported from mountain tops so far[6,7]. However, our study focused on frequent species only (i.e., species with at least 40 occurrences in the historical survey). From the total set of 759 species recorded in both the historical survey and the current re-survey data, only 135 were included in our models. Frequent species should also be those that produce largest seed yields, hence highest propagule pressure and should thus have higher-than-average colonization rates[33] and less colonization credit. It is less clear whether and how the magnitude of extinction debt is related to species frequency. However, one may speculate that this relationship is non-linear: frequent species may show higher debts because larger populations go extinct more slowly[27], and rare species may show higher debts because their long-term survival in an area probably requires above average persistence capability at the sites once occupied[34]. It hence appears plausible that disregarding the vast majority of rare species implies that our results under—rather than overestimate the current magnitude of both colonization credit and extinction debt in the Alpine flora.

The distribution of extinction events and debts, as well as colonization events and credits, along elevation is in line with a general upward movement of the mountain flora. Extinctions predominate at the lower limits of historical species ranges, colonizations at the upper limits. Extinction debts are closer to median elevations (though not significantly so), colonization credits are more distant. The pattern matches the intuitive expectation that lowest outposts tend to vanish first when rear edges retreat, and highest outposts of newly suitable sites become colonized with greatest delay when leading edges expand[35]. It is also well in line with the predominance of colonization over extinction events in high alpine monitoring plots[36] and mountain top floras[5–7], because these high-elevation sites are above the optimum elevation of the vast majority of species. Nevertheless, overlap in the elevational distribution of all four phenomena is

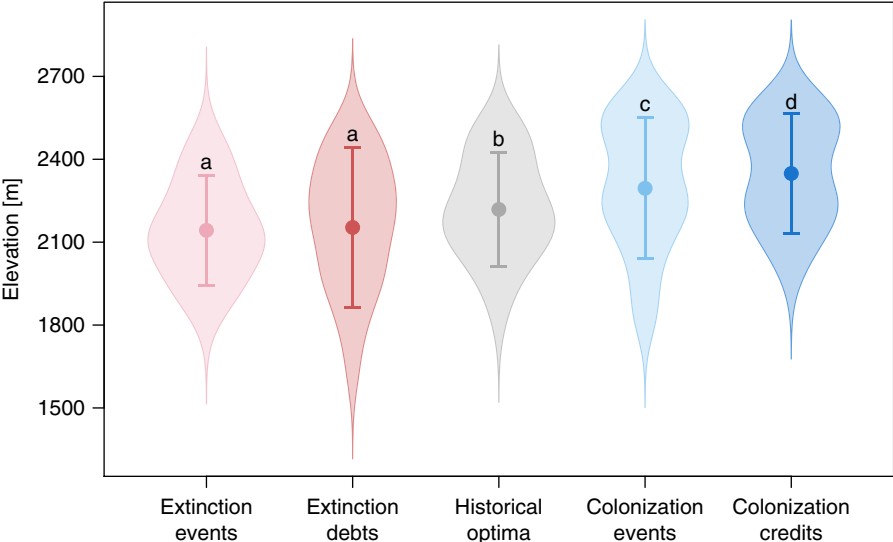

**Fig. 2** Elevational distribution of realized and unrealized extinction and colonization events. For each species, we calculated the median elevation of all plots where a species was observed historically (historical optima; gray; n = 135 species), where extinction events (light red; n = 135 species) and colonization events (light blue; n = 133 species) were observed, and where extinction debts (red; n = 128 species) and colonization credits (blue; n = 134 species) were detected. Density distributions of these median elevations per species are depicted as shaded polygons with outliers removed to improve clarity and the averages with standard deviations across all species as dots and whiskers, respectively. Letters above error bars denote significant differences in elevational positions between the five groups based on paired t-tests with Bonferroni correction. Source data are provided as a Source Data file

considerable. In particular, both extinction and colonization events span approximately the same elevational gradient as historical optimum elevations, suggesting that both of these events also occur near these optima, and likely even above them. Lateral spread, probably from warmer to colder microsites in the rugged high-mountain terrain[24,25], has hence apparently played a role in the range dynamics of the recent decades, even if it did not override the general shift towards higher elevations[22]. In particular, the preponderance of extinctions towards lower range limits indicates that retreating toward cold microsites is a strategy that may reduce, but is not able to entirely mitigate warming induced risk to mountain species[23].

The magnitude of extinction debt and colonization credit detected in the 135 species is related to several species-specific properties, even if the strength of this relationship was only moderate or low[29]. In particular, both debts and credits were related to the (historical) position of species ranges along the elevational gradient with species from higher elevations showing higher extinction debt and lower colonization credit. Relations with thermal indicator values match these patterns. Opposite relationships with elevational species niches are likely responsible for the fact that we could rarely detect both extinction debt and colonization credit for the same species. Higher levels of extinction debt in high elevation species are, again, in line with the low number of extinction events reported from high elevation monitoring sites so far[5–7]. The reasons for lower levels of extinction debt in lower elevational species may include additional pressure through changing land use, especially pasture abandonment and re-forestation[37] at lower elevations, faster competitive replacement by more vigorous plants coming up from even lower sites[10], and an increasing risk of climate warming-driven direct thermal damage with decreasing elevation[38,39]. The lower frequency of colonization credits in high-elevation species may appear intuitive at first glance, because colonization opportunities decrease towards mountain peaks. However, as we used the number of expected events to standardize credits and debts, respectively, this topographical constraint was removed from our data. Instead, we believe that the higher frequency of colonization credits in

low-elevation species is related to treeline position. Colonization credit events were most frequent for species with historical distribution peaks below 2200 m a.s.l., i.e., below the potential treeline in most parts of the study area[40]. As we only considered non-forest vegetation in our study, this implies that many of these lower elevational species are mainly growing at naturally or anthropogenically forest-free patches in a subalpine forest matrix. We hypothesize that these species show higher colonization credit because their habitats are more fragmented than those of species growing in the alpine grassland belt above the treeline, and because their seeds have to overcome forests which may represent a barrier to wind dispersal in particular. This assumption does not necessarily contradict the finding that these species have higher rates of upslope shifts at expanding range limits[22] because even species with peak distributions below the treeline usually have upper range limits above the treeline.

The idea that dispersal processes have influenced the realization of colonization opportunities during recent decades is corroborated by the relationship between seed traits indicating dispersal capability and colonization credit. Compressed climatic gradients in mountainous terrain hence do not relax dispersal limitations, at least not completely so[41]. Indeed, even in the grassland belt above the treeline many species are associated with peculiar habitats such as snow beds, wind-blown ridges, scree, or alpine fens which are discontinuously distributed across the landscape[42]. Shifting upslope hence also implies finding suitable habitat patches at these higher elevations which may become an issue even for grassland species in the upper alpine belt[36,43]. We emphasize, however, that dispersal traits only explained a minor fraction of the variation in colonization credits across the 135 species. Other factors, particularly those promoting establishment rather than dispersal may thus be even more important for the magnitude of colonization credit in mountain plants[14]. Several of them, such as seed size, are actually negatively related to dispersal capability.

The indicator of persistence capability used in our analyses was unrelated to both types of lagged population dynamics. This is surprising given that persistence capability is commonly meant to

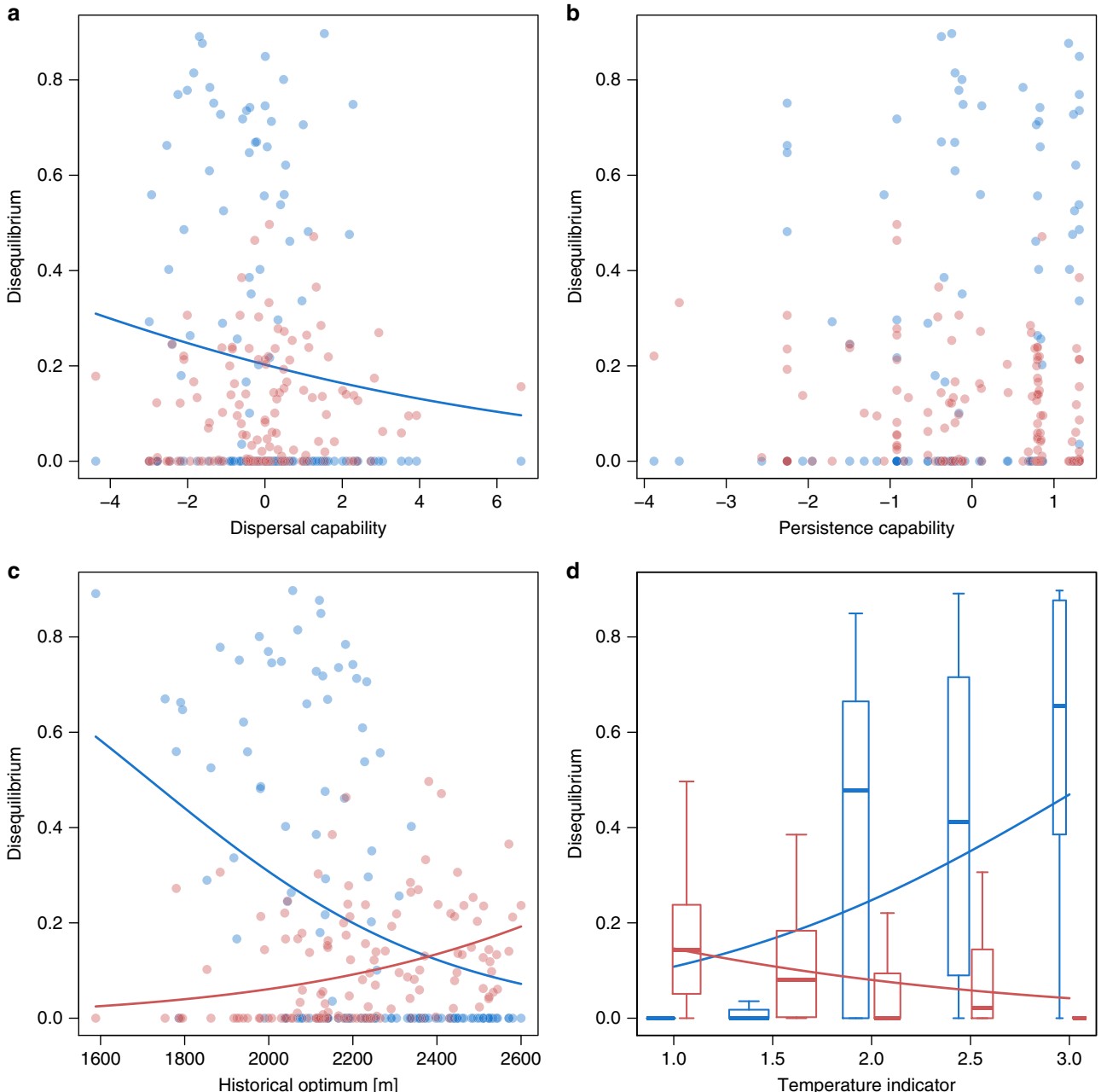

**Fig. 3** Relationships of ecological disequilibria with species-specific properties. Values of extinction debts (red) and colonization credits (blue) are relative to expected extinction and colonization events, respectively (see Methods section). Dots and boxplots represent species and are darker and wider with more species at the same position, respectively. The center line, bounds of box, and whiskers of the boxplots are defined as median, first and third quartile, and 1.5 times the interquartile range, respectively. Significant back-transformed model estimates are depicted as lines and confidence intervals are therefore not shown to improve clarity (see Supplementary Table 1 for coefficients, and Methods section for details). For dispersal capability (**a**) high values represent higher dispersal capability of species by wind and animals ($n = 134$ species), and for persistence capability (**b**) high values represent competitive, long-living and dominant species ($n = 134$ species). Historical optimum (**c**) was calculated as the median elevation of all plots where a species had been recorded in the historical data ($n = 135$ species). Temperature indicators (**d**) represent alpine to nival (1), lower alpine to upper subalpine (1.5), subalpine (2), lower subalpine to upper montane (2.5), and montane (3) species ($n = 130$ species). Source data are provided as a Source Data file

be the most important determinant of extinction debt in alpine plants[14,35], but there are several plausible reasons. First, two of the three traits used to calculate species' persistence capability were in an ordinal scale (life history strategy and dominance in the field), allowing for less dispersion of the indicator opposed to numerical traits which were used for dispersal capability. Second, intra-specific trait variability can be pronounced in plant species[44] but could not be accounted for in our analyses. And, third, even species that are able to persist for decades at sites no longer

climatically suitable for reproduction[45] may not be able to survive other environmental changes such as land use changes. For example, light-demanding grassland species are unlikely to persist when tree species are encroaching[8] even if they are competitive and long-living.

Finally, our SDMs fit (historical) realized niches in the abiotic environment, i.e., they consider the effect of biotic interactions on species distributions only implicitly[13]. Consequently, projecting these niches onto the current landscape assumes that biotic

environments have shifted in parallel with climatic conditions. For example, a site predicted to have become climatically suitable at an expanding range limit may, in reality, only have become so if it has been pre-colonized by a facilitator such as a nurse plant[46]. In fact, this assumption is unrealistic and the credits and debts have likely emerged in part because potential interaction partners have moved idiosyncratically across the landscape. To stay with the above example, colonization credits may have arisen precisely because facilitators did not pre-occupy the newly suitable sites, or, vice versa, a species may not have gone extinct from a site as predicted because the site has not been colonized by a superior competitor yet. To conclude, the simultaneous but individualistic range shifts of all species of a regional flora creates an intricate network of mutually dependent colonization and extinction dynamics. These interactions likely affect the occurrence of debts and credits as indicated by SDMs at least as much as the traits of the individual species do.

Our analyses were based on projecting historical realized niches of species onto current climatic conditions. Caveats of this approach are that geographic positions of the re-surveyed plots did not exactly match those of the historical plots, and both past and current climatic conditions at the plots were derived from statistically downscaled meteorological data. However, we standardized error probabilities during species distribution modeling and corrected all model projections by a complex calculation of error rates to reduce the potential bias to a minimum. Furthermore, both caveats apply equally to both types of disequilibrium and all species analyzed here. Therefore, we may slightly overestimate or underestimate the overall amount of disequilibrium but the balance between extinction debts and colonization credits, their relative elevational positions, and their relation with species-specific properties are likely unaffected. We emphasize, however, that colonization events may intrinsically be more difficult to predict than extinction events because apart from the demographic response to a changing climate colonization also depends on the notoriously erratic process of seed dispersal[47].

Another problem with our approach may arise if historical distributions had already been out of equilibrium with historical climates. For example, if part of the historically occupied sites had actually been unsuitable for the species, current extinction debts may appear exaggerated. However, a recent comprehensive re-survey of changes to mountain top floras across Europe suggested that from the late 19th century until the 1970s species' richness increases were detectable but of a much lower scale than during the most recent 30 years[5]. As all our plots have been first surveyed before 1970, assuming a relatively stable period with a near-equilibrium distribution of species at the time of first record hence appears reasonable[27].

Further, our analyses disregard species' adaptability, which may result in the emergence of (apparent) debts and credits. In particular, plasticity might allow populations to delay extinction under changing environmental conditions and prolong the period during which evolutionary adaptation can occur, especially at species' lower range limits[11,12]. Analogously, more plastic species can potentially colonize a wider range of suitable habitat conditions[12]. We do not know how such plasticity has affected the debts we found in this study as we lack necessary data on trait variation and adaptability for almost all species[12]. In general, adaptability, whether phenotypic, genetic or epigenetic, can certainly be important for the recent and future response of species to climate warming, but relatively little is known about its actual impact so far[11,12,48].

Taken together, our results suggest that even among the most frequent subalpine and alpine plant species of the European Alps, the realized niche of only a few species has followed climatic changes of the recent decades without delay. Even if community attributes such as average thermal indicators or species numbers respond to climatic changes quickly[1,5,6], lags in range shifts are hence obviously common among individual mountain plants. These lags are not randomly distributed across species and landscapes. From a conservation perspective, the most worrying pattern is that species from highest elevations, i.e., those with least opportunities to escape from a warming climate, are also those showing highest extinction debt. The resistance against warming that high alpine and sub-nival species have so far demonstrated on mountain tops may hence indicate that local climate has not yet changed beyond what these species are able to tolerate[43]. But, to a certain extent, this resistance apparently also masks future extinctions that have already been triggered. In such cases, assisted migration to disconnected sites at even higher mountains may represent a conservation opportunity, although such measures are under debate[49], and can hardly be implemented for more than a few species. The most powerful measure would certainly be halting climate change which might allow for evolutionary adaptation and recovery of populations before debts are paid off[11].

## Methods

**Statistical software**. All analyses were conducted in the programming environment R[50].

**Species data**. In the vegetation periods of 2014 and 2015, we re-surveyed 1516 historical plots of non-forest vegetation spread over the European Alps (Austria, Switzerland, Italy, Slovenia, and Germany) and the Swiss National park provided another 60 re-surveyed historical plots. Re-location of plots followed a standardized design that aimed at determining plot location as exactly as possible based on the metadata reported in the historical publications. At the locality described in the original publication, we first selected all topographically suitable sites based on matching a 25 × 25 m digital elevation model and a layer of substrate calcareousness with the reported values of elevation, slope inclination, exposition, and bedrock. Candidate sites were further reduced to a 200-m distance to trails, because authors of the original publications reported using trails to reach their study sites. The re-survey coordinates were then defined as the centroid of the remaining area. Precise re-survey coordinates were slightly adapted in the field based on coarse classification of habitat type (rock/scree, snow bed, meadow/heath, tall herbs, bog/swamp) and additional information about land use and topography (e.g., 'wind-blown ridge'; see Rumpf et al.[22] for further details). In total, these 1576 plots span an elevational gradient from 485 to 3226 m a.s.l. and cover time intervals of 45 to 104 years (Supplementary Fig. 2). From the total of 1070 vascular plant species recorded we excluded (i) all tree species since they were only represented by seedlings or saplings; (ii) all species that had less than 40 occurrences in the historical data set to achieve sufficiently informed species distribution models (SDMs); and (iii) all species whose historical lower range limit was not covered by our data set[22] since model parametrization is likely unreliable for these species. We did, however, not exclude species with historical upper range limits expanding above the sampled elevational range since temperature has increased in the study area between surveys and recently re-surveyed plots are thus unlikely to cover species' cold distributional limit if this was not the case historically. Additional exploratory analyses excluding these species (*Poa minor*, *Pritzelago alpina*, and *Saxifraga bryoides*) or species with restricted geographical distribution within the study area (*Gentianella aspera*, *Pedicularis aspleniifolia*, *Primula integrifolia*, *Trifolium alpinum*, and *Valeriana celtica*) confirmed our assumptions and the results were qualitatively identical. The final data set used for further analyses comprised 177 species.

**Time series of climatic data**. To represent spatial variation in temperature and precipitation as reliably as possible, we combined a high-resolution data set for precipitation of the European Alps[51] with the global temperature data set provided by Worldclim (www.worldclim.org). We downscaled average monthly precipitation sums from the 5 km resolution to 100 m using ordinary krigging with elevation as covariable, and monthly mean, minimal and maximal temperatures using the methods applied in Dullinger et al.[15]. To adjust the temperature values (averages of 1950–2000) to the time interval of the precipitation data (1970–2005) we used the E-OBS climate grids (available online www.ecad.eu/download/ensembles/download.php). The resulting grids defined the spatial pattern of climatic variation in the Alps.

To add temporal variation, i.e., to create a time series of this pattern across the entire study period, we downscaled the gridded data set CRU[52] for the Alpine arc from the original 0.5° resolution to 100 × 100 m for the years 1900–2014 by applying the change factor method[53] as used in Zimmermann et al.[54] and Dullinger et al.[15]. CRU includes minimum, maximum and mean

monthly temperatures and monthly precipitation sums. Grids of these variables were first subtracted from the monthly averages of the years 1970–2005 of the same CRU data. Subsequently, these differences were spatially interpolated to a resolution of $100 \times 100$ meters. The resulting difference grids were then subtracted or added to the spatial pattern of temperature and precipitation derived from the Isotta et al.[51] and Worldclim data as described above. Finally, we calculated thereof an annual time series of all 19 bioclimatic variables provided by Worldclim (www.worldclim.org/bioclim).

**Species distribution models.** We selected three climatic variables as predictors for the SDMs. We limited the number of variables to three to avoid model overfitting, in particular of the regression based modeling algorithms[55]. The three variables should represent the climatic conditions experienced by plants both during the vegetation period and in winter, and they should not be overly strongly correlated[56]. Based on this reasoning we chose the following variables: minimum temperature of the coldest month which is related to frost risk during winter; mean temperature of the warmest quarter which indicates available thermal energy during the vegetation period; and precipitation of the warmest quarter, related to risk of drought during the vegetation period. We note that many species are not exposed to low winter temperature because they are protected by snow. However, as snow cover is strongly determined by topography in alpine terrain[57], combining frost risk and topographical descriptors (see below) in a multivariate model is ecologically meaningful. For each plot we calculated the average of each variable for the ten years preceding the historical and recent survey, respectively (Supplementary Data 1).

To represent topography, which has as strong effect on snow accumulation, melt out times, and solar radiation income in mountainous terrain[42], we further used inclination and exposition of the plots as measured in the field (both in degrees). Finally, to account for the fact that many mountain plants are bound to either calcareous or siliceous substrates, we included a map of calcareous soils[15] (in %; Supplementary Data 1) as further predictor into the models. These three additional variables were assumed to have remained unchanged between historical sampling and re-surveys.

We parameterized SDMs within the BIOMOD2 framework[58] by relating species historical presence/absence data in the 1576 plots to the topographical and soil variables in combination with the historical values of the three climatic variables. We combined nine modeling techniques with default modeling options (see Supplementary Methods): Generalized Linear Models (GLM), Generalized Additive Models (GAM), Boosted Regression Trees (GBM), Classification Tree Analyses (CTA), Artificial Neural Networks (ANN), Surface Range Envelopes (SRE), Multiple Adaptive Regression Splines (MARS), Flexible Discriminant Analyses (FDA) and Random Forests (RF). For each technique, parameterization was replicated three times using split-sampling (80% training and 20% evaluation) with weighting for prevalence (i.e., absences and presences were weighted equally). For each replicate, model accuracy was evaluated by means of the True Skill Statistic score[59] (TSS). Based on those parameterized models reaching a TSS > 0.6 we subsequently generated ensemble projections of species distributions across all 1576 plots under both historical and recent climates. The ensemble projections were converted to presences and absences using the threshold that maximized the TSS score. As the geographic position of re-surveyed plots did not exactly match the position of the historical plots, we used a procedure similar to a k-fold cross-validation to standardize error probabilities when projecting to both historical and recent data. For each species, the data set was randomly split into ten parts, each containing 10% of the plots with equal distributions of presences and absences as in the entire data set. For each of these ten parts, a model was than calibrated with the 90% of omitted plots and projections were made onto the focal 10%. This procedure was hence repeated 10 times to produce projections of the species' presence or absence at each plot under both the current and historical climatic conditions. To avoid possible bias from splitting the data randomly into ten parts, we repeated this entire process ten times, resulting in ten presence/absence projections per plot (pair) and species. We then used a simple majority rule on the subset of these ten projections where the underlying model had reached a TSS > 0.6 to distinguish final presence and absence projections per plot, and determined it as presence in case of a tie. As a result of this procedure, we excluded 42 of the 177 species from further analyses, because the underlying models of these species never reached a TSS > 0.6. Comparing historical field observations and final historical model projections resulted in a mean sensitivity of 0.74, a mean specificity of 0.87 and a mean TSS of 0.61 across the remaining 135 species (Supplementary Data 2).

**Calculating colonization credit and extinction debt.** We heuristically combined field observations and model projections to assess whether species' current distributions are in disequilibrium with climatic conditions in the following way: if a species was observed during the historical survey in a particular plot but was projected by the model to be absent from the respective paired plot under current climatic conditions we defined this constellation as an 'expected extinction event'. If the species was nevertheless observed in this plot during the re-survey we counted this case as an 'extinction debt event'. Vice versa, if a species was not observed in a particular plot during the historical survey but was predicted by the model to be present under current climatic conditions, we defined this

constellation as an 'expected colonization event'. If the species was nevertheless not observed in the paired plot during the re-survey we counted this case as a 'colonization credit event'.

This heuristic is hence based on differences between field observations and model projections under the climatic conditions of the recent re-survey. However, such differences might not only arise from (delayed) population dynamics but also from sampling errors and errors of omission (false absences—potentially confounded with extinction debt) and commission (false presences—potentially confounded with colonization credit) of the SDM. To account for these errors when calculating extinction debts, we subtracted from the number of cases of extinction debt events, as defined above, the number of cases likely attributable to omission errors as evaluated on the basis of the historical data. That is, we computed for each species:

$$P_{H_{1x} \to R_{10}} - \left( P_{H_{1x} \to R_{1x}} \times \left( \frac{P_{H_{10}}}{P_{H_{1x}}} \right) \right), \quad (1)$$

where $P_{Hij \to Rij}$ represents the number of plots ($P$) that were historically ($H$) and recently ($R$) observed in state $i$ and predicted in state $j$. The states ($ij$) indicate the presence (1), absence (0) or either presence or absence ($x$) of a given species in that plot. $P_{H1x \to R10}$ are hence all predicted extinction events that did not occur, i.e., all extinction debt events as defined above. $P_{H1x \to R1x}$ are all plots where the species was observed both historically and in the re-survey, irrespective of model predictions, hence the theoretical maximum of extinction debt events; and $P_{H10}/P_{H1x}$ is the omission error of the historical model predictions, i.e., the number of incorrectly predicted absences divided by the number of observed presences. To put it differently, the term in brackets computes the number of apparent extinction debt events attributable to lack of model accuracy.

Analogously, when calculating colonization credits, we subtracted from all cases of colonization credit events, as defined above, the cases attributable to commission errors as evaluated on the basis of the historical data. That is, we computed:

$$P_{H_{0x} \to R_{01}} - \left( P_{H_{0x} \to R_{0x}} \times \left( \frac{P_{H_{01}}}{P_{H_{0x}}} \right) \right), \quad (2)$$

where the terms are analogous to Eq. (1) and $P_{H01}/P_{H0x}$ hence represents the commission error of the historical model predictions, i.e., the number of incorrectly predicted presences divided by the number of observed absences.

To standardize the numbers of extinction debt and colonization credit events across species with unequal prevalence and number of suitable sites in the dataset, we divided the corrected number of cases of extinction debt and colonization credit by the number of expected extinction and colonization events for each species, respectively. The number of expected extinction events is equal to the sum of all extinction debt events ($P_{H1x \to R10}$) and all correctly predicted extinction events ($P_{H1x \to R00}$), both of which include a certain error rate. As stated above, we subtracted from the originally predicted number of extinction debt events the number of cases likely attributable to omission errors (Eq. (1)) which should thus be regarded as cases of persistence ($P_{H1x \to R11}$). Following the same logic, we further calculated the number of cases where unexpected extinction events ($P_{H1x \to R01}$) are likely due to commission errors and should thus be added to the correctly predicted extinction events ($P_{H1x \to R00}$) as:

$$P_{H_{1x} \to R_{00}} + \left( P_{H_{1x} \to R_{0x}} \times \left( \frac{P_{H_{01}}}{P_{H_{0x}}} \right) \right) \quad (3)$$

and used this corrected value when standardizing across species with unequal prevalence and number of suitable sites (Supplementary Fig. 1).

Vice versa, the number of expected colonization events is equal to the sum of all colonization credit events ($P_{H0x \to R01}$) and correctly predicted colonization events ($P_{H0x \to R11}$). Thus, we calculated the number of cases where unexpected colonization events ($P_{H0x \to R10}$) are likely due to omission errors and added them to the number of correctly predicted colonization events as:

$$P_{H_{0x} \to R_{11}} + \left( P_{H_{0x} \to R_{1x}} \times \left( \frac{P_{H_{10}}}{P_{H_{1x}}} \right) \right) \quad (4)$$

and, again, used this corrected number for standardization (Supplementary Fig. 1).

**Species-specific properties.** Based on 4460 values of own measurements, as well as literature and online data bases (see Supplementary Table 1 for references) we calculated the means of the four properties dispersal capability, persistence capability, historical optimum elevation, and temperature indicator value for each species. To indicate species' dispersal capability we first estimated retention times of seeds in fur of cattle, deer, and rabbit, as well as in fur of horse, sheep, and bear, and seed survival in animal guts based on the traits seed surface structure and seed mass following Römermann et al.[60] and Mouissie et al.[61]. We then calculated a principal component analysis using the function *dudi.pca* as implemented in the package *ade4*[62] of the standardized values of these three estimates of zoochorous dispersal (i.e., two types of retention times, and seed survival), as well as the release height and the terminal falling velocity of the species' seeds—two traits important for wind dispersal in particular[63]. One species (*Huperzia selago*) was removed from this analysis as an extreme outlier and missing values of these traits (68 out of

670 values) were imputed prior to the principal component analysis using the function *imputePCA* as implemented in the package *missMDA*[64]. Species' values along the first axis of this ordination were used as indicator of dispersal capability, where high values represent higher dispersal capability of species by wind and animals. As index of a species' capability to persist in habitats no longer climatically suitable we used values of the first axis of a correspondence analysis, as implemented in the function *dudi.mix* of the package *ade4*[62], based on data on species' life history strategy (competitive, stress-tolerant, mixed), life span (years) and dominance in the field (dominant, scattered). Again, one species (*Juniperus communis*) was removed as an extreme outlier and missing values (49 out of 405 values) were imputed prior to the correspondence analysis using the function *imputeFAMD* from the package *missMDA*[64]. High values represent competitive, long-living and dominant species. Species' historical optimum elevation was calculated as the median elevation of all plots where a species had been recorded in the historical data. Species' temperature indicator values were extracted from Landolt et al.[65] with three species missing information. These values indicate how thermophilic a plant species is and range from one (cryophilic, high alpine species) to five (thermophilic, low elevation species). Only two species had temperature indicator values greater than three (i.e., 3.5 and 4) and were thus removed from the analyses as outliers.

**Statistical analyses**. Whether extinction debts and colonization credits of the plant species were significantly different from each other was assessed by a Wilcoxon signed-rank test using the function *wilcox.test*, and the Pearson's correlation coefficient of extinction debts and colonization credits was calculated using the function *cor*. For analyzing how disequilibria were distributed across the elevational ranges of species, we calculated, in addition to historical optimum elevations, the median elevation of all plots per species where extinction and colonization events, as well as extinction debt and colonization credit events occurred. Whether the medians of these five types of elevational positions differed from each other across species was then assessed by paired *t*-tests with Bonferroni correction for multiple tests using the function *pairwise.t.test*. For this analysis we did not correct model predictions for omission and commission errors since it is impossible to distinguish cases of extinction debt or colonization credit, respectively, from model-specific errors on a plot basis. Moreover, the focus here was not on the number of predicted events, debts and credits but on their relative elevational positions.

For assessing the relations of disequilibria with species-specific properties we applied linear regressions for beta-distributed data using the function *betareg* with a logit link as implemented in the homonymous package[66] with either extinction debt or colonization credit as response and each species-specific property as predictor variable in individual models. However, several of the species in our data set had an extinction debt or colonization credit of zero. Therefore, we first transformed the data following Smithson and Verkuilen[67] with the formula $(y^{\star}(n-1)+0.5)/n$, where $y$ corresponds to either extinction debt or colonization credit, and $n$ to the number of species (i.e., 135). This transformation hence squeezes all values minimally towards 0.5 and is reasonable for our data since it is very likely that a case of extinction debt or colonization credit would be observed at least once if the sampling size increased. Normal distribution of residuals and homogeneity of variances was verified visually for all models.

**Reporting summary**. Further information on research design is available in the Nature Research Reporting Summary linked to this article.

## Data availability

Re-survey data are available online in the Phaidra database at https://phaidra.univie.ac.at/view/o:630655. The source data underlying Figs. 1–3 are provided as a Source Data file. Climatic data and values of calcareousness used for species distribution modeling are provided in Supplementary Data 1. Trait values used to calculate species' dispersal and persistence capability are partly derived from the online database TRY[68]. Following the intellectual property guidelines of TRY, raw trait data are not publicly available but can be obtained from the authors upon request and consent from TRY.

## Code availability

Please see Supplementary Methods for species distribution model calibration, ensemble modeling, and projection. Further code details can be obtained from the authors upon request.

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

## Acknowledgements

We thank M. Schütz for the contribution of re-survey data; the Swiss National Park and the National Park Hohe Tauern for their cooperation; C. Kuehs, S. Ertl, A. Dellinger, M. Sonnleitner, N. Helm, S. Stifter, E. Kucs, N. Sauberer, and C. Gilli for their help in the field; B. Kramer for the use of the velocimeter; and S. Venn, K. Steinbauer, G. Gimpl, and M. Stehlik for the contribution of trait data. This study was funded by the Austrian Climate Research Program, project number B368575, and a Dissertation Completion Fellowship of the University of Vienna. Open access funding was provided by University of Vienna. The study has been supported by the TRY initiative on plant traits (http://www.try-db.org). The TRY initiative and database is hosted, developed and maintained by J. Kattge and G. Bönisch (Max Planck Institute for Biogeochemistry, Jena, Germany). TRY is currently supported by DIVERSITAS/Future Earth and the German Centre for Integrative Biodiversity Research (iDiv) Halle-Jena-Leipzig.

## Author contributions

S.B.R., K.H. and S.D. designed the study; S.B.R., K.H., W.W. and S.D. conducted field-work; W.W. contributed data; J.W., D.M., A.G. and G.K. downscaled climatic data; S.B.R. analyzed the data with input from K.H., J.W. and S.D.; S.B.R. led the writing with substantial contributions from N.E.Z. and all other authors.

## Additional information

**Competing interests:** The authors declare no competing interests.

