## [Peer Review File · Nature Communications]

Reviewers' comments:

Reviewer #1 (Remarks to the Author):

This study aims to detect "possible disequilibrium dynamics" in the non-forest vegetation of the European Alps during the recent decades by using the results of species distribution modelling with those of presence/absence data of the recent re-survey of historical plots. In particular, authors ask i) whether the current distribution of alpine plants in the study area matches the distribution from their historical climatic niches, ii) whether mismatches are due to unrealized extinction or colonization events, and iii) whether there is an elevation al structure in the frequency of such unrealized events, where they expected extinctions debts being more prominent at lower range limits and colonization credits at the upper range limit. In my opinion this is very interesting paper that provide important information about the potential disequilibrium that might be occurring between the presence of some species in a particular site or community and the abiotic conditions that they are currently facing. The methodology is in general rigorous and the analyses robust. The text is clear, and straightforward. However, I have some concerns about the used approach that require some clarification and to be discussed in the text.

First, the SDM used only abiotic variables that were downscaled to account for the fine-grain of the re-survey data. In general, but for alpine plants in particular it has been shown the biotic factors are very important for explaining their distribution. For instance, the presence of positive interactions among species, which are expected to be more frequent in alpine species, imply that the presence of other species (i.e. nurse) is the key factor for explaining the presence of one or several species at given site or community. In my opinion this issue should be addressed, or at least discussed, by the authors.

Second, why authors only used 3 abiotic variables for the SDM? More importantly, I don't agree with the reasoning for the use of the minimum temperature of the coldest month as in the study area snow cover protect several plants from freezing damage. Thus, a better explanation should be indicated.

Finally, there is an important factor that should also be considered in the explanation and discussion of the results: species plasticity. The approach used do not consider this possibility that might play an important role in the final set of extinction debts and colonization credits. Authors should be aware that the spatial distribution of a species doesn't necessarily reflect its plasticity. Minor points.

L41-44. Another possibility is that models are somehow inaccurate as they can miss the inclusion of some important biotic factor driving plant species distribution.

L50-51. Or both?

L61. I don't agree that balance is important. Are we ok if the amount of species with extinction debt is the same amount of those with colonization credit?, or is more important more credits than debts? or vice-versa?

L93-94. Any citation to support this?

L438-452. It is not totally clear to me how species with wind dispersal were handled.

Reviewer #2 (Remarks to the Author):

I think this is a really useful and novel paper that marries two types of approaches to produce an important look at how species have been responding to climate change and the degree to which lags in both colonization and extinction might exist. I don't have any giant concerns, but think there are a few things that should be done to improve the paper (these are listed in no particular order).

In the introductory paragraph the ms states "...results hence suggest that almost no species is able to fully track climate change". This statement is made in the context of both extinction debts and colonization credits. This makes sense in the context of colonization credits. However, in the

context of extinction debts, this doesn't make sense if the trailing edge of a shifting distribution is set by biotic interactions (and not by climate). If that's the case, it is not that the species is not tracking climate, its just that the climate space its "tracking" is larger than its historic niche, because historically, its lower elevation range boundary was set by something other than climate. This isn't a big deal, but I think a slightly more nuanced message here would be better.

The authors stated: "However, if extinction debts occurred, they were, on average, smaller (16% of expected extinction events) than colonization credits (54% of the expected colonization events)." I didn't understand what this meant. Where they saying something about the extent of expected elevation change? This needs to be clarified, both here and in other spots in the ms where they make these sorts of statements.

Fig. 2 – the figure legend is not sufficient, as its hard to know what's being shown in this figure without reading the main text. Adding clarifying text to the legend would be helpful.

Fig. 3d – these patterns are expected from a null expectation. There is no where higher for the highest elevation species to go, so they must (by definition) have an immigration credit score of zero, whereas the opposite is true for low elevation species. So, you should expect to see the relationships shown as "significant" in the panel. You could run simulation models and see if the slopes are different than you'd expect by chance, it would be interesting to know if anything was happening more slowly or quickly then expected by chance.

Note that Fig. 3c. suffers from the same null expectation concerns and similar solutions could help to better contextualize this result.

The ms states: "Eighty-three percent of the plant species in our data set had either only extinction debts or only colonization credits, while only 10% had both types of disequilibrium (inset Fig. 1)." This is a bit misleading for the same reasons described above, namely that some percentage of the species by definition can't have an immigration credit, because they are already at the highest elevations considered. So, I think these numbers need to better contextualized or perhaps reported differently.

Reviewer #3 (Remarks to the Author):

REVIEWER COMMENTS - NCOMMS-19-16744

The authors analysed extinction debt and colonization credit for 135 non-forest plant species in the European Alps. They tested whether current distributions matched those expected from their historical climatic niches and whether mismatches are likely to be due to (as yet) unrealized extinction or colonization events. They calculated extinction debt in 60% of species and colonization credit in 38% of species (with at least one effect in 93% of species). Colonization credits were more frequent in thermophilic species from lower elevations and conversely extinction debt was more common among cold-adapted species from highest elevations. The study is likely to be of interest to others studying mountain ecosystems and to be influential as most previous studies have assessed communities rather than individual species. By examining the past in relation to the present this study complements an earlier study with some of the same co-authors (Dullinger et al. 2012) that compared present distributions with projections for 2100.

Title: Consider a slightly more informative title – Extinction debts and colonization credits in non-forest plant distributions in the European Alps

Keywords: okay

The introductory, results, discussion and methods sections are generally fine. My comments below relate mainly to minor editorial suggestions.

Use of species distribution models and statistical analyses are appropriate.

Figures are appropriate. Figure 2 provides a particularly useful summary.

Supplementary information is appropriate and adequate.

The Reporting Summary provides a useful overview of the whole analysis.

Recommendation: Accept following minor modifications.

Page 3 Line 37 – ..many new species have been colonizing mountain tops.. many species previously found at lower elevations.. rather than 'new'.

Page 3 line 43 Delete 'can usually not' and insert 'cannot usually'... delete 'about'

Page 3 Line 47 Surely there are already species in disequilibrium with current conditions? Such disequilibria are not just expected in the future. In fact, you write later on

page 7 line 142 'disequilibria are currently widespread among mountain plants'.

Page 3 Line 48 Shouldn't this be "whether such disequilibria have already emerged in mountain plant distributions..."?

Page 3 Line 56 Delete 'or' and insert 'and'

Page 4 Line 66 Delete 'similar' and insert 'in a similar way'

Page 4 Line 74 Delete 'the recent' and insert 'recent'

Page 4 Line 77 Delete 'thereby' and insert 'therefore'

Results - Well summarized – no particular problems.

Discussion

It could be worth stating somewhere that colonization credit is likely to be more difficult (possibly much more difficult) to predict than extinction debt. Extinction debt focusses on one species at a time (i.e. can it survive, but not reproduce under certain future conditions where it is already located), whereas colonization debt requires a minimum of information about at least two species and possibly many more (i.e. can the colonizer reach a 'new' site and will the one or more species presently there not be there at some future time). Estimating colonization debt also involves some challenging assumptions e.g. dispersal assumptions mentioned on page 19 lines 442-3.

Page 8 Line 159 Delete 'frequent species' and insert 'species which occur frequently in the dataset' or whatever is appropriate. Do you mean species with more than 40 occurrences as stated on Page 13 line 294?

Page 9 Line 198 Delete 'is' and insert 'are'

Page 10 Line 221 Delete 'the'

Page 10 Line 224 Delete 'bound to' and insert 'associated with'

Page 10 Line 230 Delete 'hence' and insert 'thus'

Page 12 Line 261 Delete '1970ies' and insert '1970s'

Page 12 Line 270 Delete 'hence'

Page 12 Line 277 Delete 'it' and insert 'this resistance'

Methods

Page 13 line 288 (and Page 11 Line 248) Please clarify exactly what you mean by 're-located'. I assume you mean you located as best you could the historical plots, perhaps from old maps or rough pre-GPS lats and longs? Add some phrase similar to that you have in the Reporting Summary i.e. as the geographic position of re-surveyed plots did not exactly match the position of the historical plots...

Page 14 Line 315 Delete 'thus created' and insert 'resulting'

Page 15 Line 349 I don't require a reanalysis, as nine SDM techniques are used in an ensemble model. However, I do question the use of just three climatic variables and three additional variables with the SRE/BIOCLIM approach. The original BIOCLIM SDM program used 12 variables in 1984 and produced useful results. However, these were increased in several stages to 35 variables in 1999. As the SRE/BIOCLIM approach is very simple (i.e. just range or percentile range) it requires many variables to work effectively. When used correctly it produces results almost as good as Maxent (see Booth 2018 'Species distribution modelling tools and databases to assist managing forests under climate change' and especially the maps in the Penman et al (2010) reference given there). I use and recommend Maxent, but the BIOCLIM approach is much better than many comparisons suggest.

Page 16 Line 362 Delete 'the left out', delete 'plots' and insert 'plots omitted'

Page 18 Line 415 Delete 'so'

Page 18 line 417 Delete 'Yet,'

Page 19 Line 438 Delete 'from' and insert 'of', delete 'measurements' and insert 'measurements as well as'

Reviewer #1 (Remarks to the Author):

This study aims to detect “possible disequilibrium dynamics” in the non-forest vegetation of the European Alps during the recent decades by using the results of species distribution modelling with those of presence/absence data of the recent re-survey of historical plots. In particular, authors asked i) whether the current distribution of alpine plants in the study area matches the distribution from their historical climatic niches, ii) whether mismatches are due to unrealized extinction or colonization events, and iii) whether there is an elevation structure in the frequency of such unrealized events, where they expected extinction debts being more prominent at lower range limits and colonization credits at the upper range limit. In my opinion this is a very interesting paper that provides important information about the potential disequilibrium that might be occurring between the presence of some species in a particular site or community and the abiotic conditions that they are currently facing. The methodology is in general rigorous and the analyses robust. The text is clear, and straightforward.

Response: *Thanks a lot!*

However, I have some concerns about the used approach that require some clarification and to be discussed in the text. First, the SDM used only abiotic variables that were downscaled to account for the fine-grain of the re-survey data. In general, but for alpine plants in particular it has been shown the biotic factors are very important for explaining their distribution. For instance, the presence of positive interactions among species, which are expected to be more frequent in alpine species, imply that the presence of other species (i.e. nurse) is the key factor for explaining the presence of one or several species at given site or community. In my opinion this issue should be addressed, or at least discussed, by the authors.

Response: *We agree with the reviewer that biotic interactions are certainly important in alpine environments, both for the current distribution and for re-distribution dynamics under climate change. We did not incorporate biotic interactions into our models, (1) because of the generic and yet unresolved problem of how to parameterise their effects on species distributions properly, and (2) because we believe that they are one of the factors determining the emergence and extent of the disequilibria that we wanted to investigate. To highlight the latter point we have introduced an additional paragraph to the Discussion (lines 252-266). We moreover mention the crucial role of biotic interactions now also at several other places in the manuscript.*

Second, why authors only used 3 abiotic variables for the SDM? More importantly, I don't agree with the reasoning for the use of the minimum temperature of the coldest month as in the study area snow cover protects several plants from freezing damage. Thus, a better explanation should be indicated.

Response: *In particular for regression based modelling techniques, the relationship between the number of predictor variables and sample size (which is the number of occurrences in case of logistic models) is important to avoid model overfit (e.g. Harrell 2001). The general recommendation is to keep the ratio above 1:10 (e.g. Babyak 2004). As a corollary the number of predictors we could use was limited by the number of occurrences of the species. As it was important for us to base our work and conclusions on a sufficiently large set of species, we decided to include species that have at least 40 occurrences and to go for an already low ratio of approximately 1:8, i.e. six predictor variables (note that the majority of the*

177 species has > 40 occurrences and hence the ratio is higher for them). As we had to include basic information about bedrock and topography (slope orientation and inclination; terrain surface is not well represented by the downscaled climatic variables), we hence had to limit the number of climatic variables to a maximum of three.

Selecting these three variables from the overall set of 19 available in BIOCLIM was under two constraints: first, we wanted to represent both temperature and precipitation, and second we wanted to avoid highly correlated variables (which can distort regression coefficients and future projections, cf. Dormann et al. 2013 for an extensive treatment of the topic). Among temperature variables we thought, and still think, it is useful to select one that represents summer and one that represents winter conditions. Although we completely agree with the referee's argument that many sites are covered by snow during peak winter and plants hence not exposed to coldest temperatures, this is not the case for all habitats (e.g. windblown areas). As snow cover is closely correlated with topography, we thus think that the combination of both frost risk and topography in a model does make sense.

Equally importantly, alternative temperature variables that we could have used instead of temperature of the coldest month are either highly correlated with temperature of the warmest month, the other temperature variable in the model (namely: mean annual temperature, Pearson $r > 0.9$), or are ecologically equivalent to temperature of the coldest month (temperature of the coldest quarter), or are, according to own experience and trials, hardly related to mountain plant distribution in the Alps (variables representing daily or seasonal variability of temperature like isothermality). As a consequence, we still consider temperature of the coldest month the most reasonable choice if it is, as in our models, combined with topography and temperature of the warmest month.

In the paper, we now try to better explain and justify our choice of variables (see lines 367-379).

Finally, there is an important factor that should also be considered in the explanation and discussion of the results: species plasticity. The approach used do not consider this possibility that might play an important role in the final set of extinction debts and colonization credits. Authors should be aware that the spatial distribution of a species doesn't necessarily reflect its plasticity.

Response: We fully agree with the referee. This aspect is potentially important but insufficiently studied so far. We have added a respective paragraph to the Discussion now (lines 289-298).

Minor points.

L41-44. Another possibility is that models are somehow inaccurate as they can miss the inclusion of some important biotic factor driving plant species distribution.

Response: You are right and we agree (see above). We added a respective sentence at this place and moreover discuss this topic now in the Discussion (lines 43-45 and 252-266).

L50-51. Or both?

Response: We are not quite sure what the reviewer means here. We state that both processes are very important. We deleted the comma before 'and' and hope that this made the sentence clearer.

L61. I don't agree that balance is important. Are we ok if the amount of species with extinction debt is the same amount of those with colonization credit?, or is more important more credits than debts? or vice-versa?

Response: *We deleted the statement about the balance between debts and credits because of course both are important.*

L93-94. Any citation to support this?

Response: *The given citation was meant for all three named drivers. We changed the structure of the sentence to clarify this.*

L438-452. It is not totally clear to me how species with wind dispersal were handled.

Response: *Wind dispersal capacity was accounted for by release height and terminal falling velocity of seeds, two traits that are decisive for dispersal distances by wind as shown e.g. by mechanistic modelling based of turbulent scalar transport (Katul et al. 2005). The respective explanation is on lines 489-493 and lines 496-498 of the revised manuscript.*

References:

- Harrell, F.E. 2001. *Regression modelling strategies*. Springer, New York.
- Babyak, M.A. 2004. *What you see may not be what you get: A brief, nontechnical introduction to overfitting in regression-type models*. *Psychosomatic Medicine* 66: 411-421.
- Dormann, C. F. et al. 2013. *Collinearity: a review of methods to deal with it and a simulation study evaluating their performance*. *Ecography* 36, 27-46.
- Katul, G. G. et al. 2005. *Mechanistic Analytical Models for Long-Distance Seed Dispersal by Wind*. *The American Naturalist* 166, 368-381.

Reviewer #2 (Remarks to the Author):

I think this is a really useful and novel paper that marries two types of approaches to produce an important look at how species have been responding to climate change and the degree to which lags in both colonization and extinction might exist. I don't have any giant concerns, but think there are a few things that should be done to improve the paper (these are listed in no particular order).

Response: *Thanks a lot!*

In the introductory paragraph the ms states "...results hence suggest that almost no species is able to fully track climate change". This statement is made in the context of both extinction debts and colonization credits. This makes sense in the context of colonization credits. However, in the context of extinction debts, this doesn't make sense if the trailing edge of a shifting distribution is set by biotic interactions (and not by climate). If that's the case, it is not that the species is not tracking climate, its just that the climate space its "tracking" is larger than its historic niche, because historically, its lower elevation range boundary was set by something other than climate. This isn't a big deal, but I think a slightly more nuanced message here would be better.

Response: *We agree and toned our statement down and now do not write that the species do not fully track climate change, but that their realized niches do not fully track climate*

change. We moreover introduced a new paragraph in the Discussion, also in response to a comment of referee #1, where we discuss the relationship of realized niches, range shifts and biotic interactions (lines 252-266).

The authors stated: “However, if extinction debts occurred, they were, on average, smaller (16% of expected extinction events) than colonization credits (54% of the expected colonization events).” I didn’t understand what this meant. Where they saying something about the extent of expected elevation change? This needs to be clarified, both here and in other spots in the ms where they make these sorts of statements.

Response: *As the Methods follows the Results and Discussion sections we agree that the reader indeed misses some important pieces of information to understand the Results (if she or he does not reverse this sequence when reading). We therefore now added a short summary of the basic logic of our approach to the Introduction (lines 85-92). There we explain that we mean colonization and extinction events predicted to have occurred on our 1576 plots between first survey and re-survey by the species distribution models when we talk about “expected colonization” or “expected extinction”. And further that colonization credit and extinction debt events are the subsets of predicted colonization and extinction events that have not been observed during the re-survey of the plots in 2014/15. We hope that with this information given in the Introduction, statements like the one cited by the Referee are now clear to the reader of the Results (without having to go into the Methods first).*

Fig. 2 – the figure legend is not sufficient, as its hard to know what’s being shown in this figure without reading the main text. Adding clarifying text to the legend would be helpful.

Response: *We agree and expanded the figure legend and explain now in more detail what is shown.*

Fig. 3d – these patterns are expected from a null expectation. There is no where higher for the highest elevation species to go, so they must (by definition) have an immigration credit score of zero, whereas the opposite is true for low elevation species. So, you should expect to see the relationships shown as “significant” in the panel. You could run simulation models and see if the slopes are different than you’d expect by chance, it would be interesting to know if anything was happening more slowly or quickly then expected by chance. Note that Fig. 3c. suffers from the same null expectation concerns and similar solutions could help to better contextualize this result.

Response: *We agree with the referee that when species approach the highest elevations further upslope colonization opportunities diminish. However, we were aware of this problem and have already accounted for it by standardizing the number of colonization credit and extinction debt events by the number of expected colonization and extinction events (see Methods lines 458-480, and captions of Fig. 1 and 3). The curves in Fig. 3d hence represent proportions of expected events and not absolute number of events. We also note that all species had expected colonization and extinction events and almost all of them more than 10 events of each kind – we now explicitly state this in the Results section (lines 107-109) and illustrate it by an additional figure added to the Supplementary Material (Supplementary Fig. 1). Hence the theoretical problem that species have no colonization credit because there is no expected colonization does not occur in our 135 species. Finally, we also note that we re-ran our analyses excluding those three species whose historical leading edges were not well within our elevational sampling range, and this yielded qualitatively identical results (cf. lines*

334-341 of the Methods section). In the paper, we emphasize this correction again in the Discussion section (lines 211-215).

The ms states: “Eighty-three percent of the plant species in our data set had either only extinction debts or only colonization credits, while only 10% had both types of disequilibrium (inset Fig. 1).” This is a bit misleading for the same reasons described above, namely that some percentage of the species by definition can’t have an immigration credit, because they are already at the highest elevations considered. So, I think these numbers need to be better contextualized or perhaps reported differently.

Response: We hope that the changes in the revised version of the manuscript and our explanations above convince the reviewer that our methods have appropriately accounted for this potential problem.

Reviewer #3 (Remarks to the Author):

The authors analysed extinction debt and colonization credit for 135 non-forest plant species in the European Alps. They tested whether current distributions matched those expected from their historical climatic niches and whether mismatches are likely to be due to (as yet) unrealized extinction or colonization events. They calculated extinction debt in 60% of species and colonization credit in 38% of species (with at least one effect in 93% of species). Colonization credits were more frequent in thermophilic species from lower elevations and conversely extinction debt was more common among cold-adapted species from highest elevations. The study is likely to be of interest to others studying mountain ecosystems and to be influential as most previous studies have assessed communities rather than individual species. By examining the past in relation to the present this study complements an earlier study with some of the same co-authors (Dullinger et al. 2012) that compared present distributions with projections for 2100.

Response: Thanks a lot!

Title: Consider a slightly more informative title – Extinction debts and colonization credits in non-forest plant distributions in the European Alps

Response: We changed the title as suggested but excluded ‘distributions’ to make it more compact.

Keywords: okay

The introductory, results, discussion and methods sections are generally fine. My comments below relate mainly to minor editorial suggestions.

Use of species distribution models and statistical analyses are appropriate.

Figures are appropriate. Figure 2 provides a particularly useful summary.

Supplementary information is appropriate and adequate.

The Reporting Summary provides a useful overview of the whole analysis.

Recommendation: Accept following minor modifications.

Response: Thank you!

Page 3 Line 37 – ..many new species have been colonizing mountain tops.. many species previously found at lower elevations.. rather than ‘new’.

Page 3 line 43 Delete ‘can usually not’ and insert ‘cannot usually’... delete ‘about’

Response: We changed both.

Page 3 Line 47 Surely there are already species in disequilibrium with current conditions? Such disequilibria are not just expected in the future. In fact, you write later on page 7 line 142 ‘disequilibria are currently widespread among mountain plants’.

Response: We fully agree. Actually, the expectation that at least some species are already in disequilibrium with current climatic conditions was the very motivation for this study. In the Introduction, we introduce, however, what has been published and discussed as yet, and this sentence refers to a well cited opinion paper (Svenning and Sandel, 2013) that exactly makes the referred claim.

Page 3 Line 48 Shouldn't this be “whether such disequilibria have already emerged in mountain plant distributions...”?

Page 3 Line 56 Delete ‘or’ and insert ‘and’

Page 4 Line 66 Delete ‘similar’ and insert ‘in a similar way’

Page 4 Line 74 Delete ‘the recent’ and insert ‘recent’

Page 4 Line 77 Delete ‘thereby’ and insert ‘therefore’

Response: We changed all five as suggested.

Results - Well summarized – no particular problems.

Response: Thank you!

Discussion

It could be worth stating somewhere that colonization credit is likely to be more difficult (possibly much more difficult) to predict than extinction debt. Extinction debt focusses on one species at a time (i.e. can it survive, but not reproduce under certain future conditions where it is already located), whereas colonization debt requires a minimum of information about at least two species and possibly many more (i.e. can the colonizer reach a ‘new’ site and will the one or more species presently there not be there at some future time). Estimating colonization debt also involves some challenging assumptions e.g. dispersal assumptions mentioned on page 19 lines 442-3.

Response: We agree, colonization is intrinsically more difficult to predict as it requires additional information about the distribution of seed sources and depends on the notoriously erratic process of seed dispersal. We now emphasize this point in the Discussion (lines 277-280).

Page 8 Line 159 Delete 'frequent species' and insert 'species which occur frequently in the dataset' or whatever is appropriate. Do you mean species with more than 40 occurrences as stated on Page 13 line 294?

Page 9 Line 198 Delete 'is' and insert 'are'

Page 10 Line 221 Delete 'the'

Page 10 Line 224 Delete 'bound to' and insert 'associated with'

Page 10 Line 230 Delete 'hence' and insert 'thus'

Page 12 Line 261 Delete '1970ies' and insert '1970s'

Page 12 Line 270 Delete 'hence'

Page 12 Line 277 Delete 'it' and insert 'this resistance'

Response: *We changed all as suggested.*

Methods

Page 13 line 288 (and Page 11 Line 248) Please clarify exactly what you mean by 're-located'. I assume you mean you located as best you could the historical plots, perhaps from old maps or rough pre-GPS lats and longs? Add some phrase similar to that you have in the Reporting Summary i.e. as the geographic position of re-surveyed plots did not exactly match the position of the historical plots...

Response: *We amended the sentence and explain the re-location now in more detail in the Methods section (lines 323-327).*

Page 14 Line 315 Delete 'thus created' and insert 'resulting'

Response: *We changed it.*

Page 15 Line 349 I don't require a reanalysis, as nine SDM techniques are used in an ensemble model. However, I do question the use of just three climatic variables and three additional variables with the SRE/BIOCLIM approach. The original BIOCLIM SDM program used 12 variables in 1984 and produced useful results. However, these were increased in several stages to 35 variables in 1999. As the SRE/BIOCLIM approach is very simple (i.e. just range or percentile range) it requires many variables to work effectively. When used correctly it produces results almost as good as Maxent (see Booth 2018 'Species distribution modelling tools and databases to assist managing forests under climate change' and especially the maps in the Penman et al (2010) reference given there). I use and recommend Maxent, but the BIOCLIM approach is much better than many comparisons suggest.

Response: *We agree with the referee that the optimal number of predictor variables varies among the nine techniques we have used. While for regression-based methods like GLM and GAM overfitting is an issue which sets an upper limit to predictors, other methods are largely insensitive to low sample size : predictor ratios (such as RF and GBM) or even work better with large predictor sets (such as SRE). However, here we wanted to apply a standard set of predictors across all nine techniques and hence limited the number of predictors according to requirements for regression based methods. This has probably weakened the*

contribution of SRE to final consensus models, but as there were eight alternative algorithms this problem is, as indicated by the referee, probably a minor one. We also note that we applied a conservative TSS-threshold of 0.6 for all models implying that final results are only based on models that received high evaluation scores.

Page 16 Line 362 Delete 'the left out', delete 'plots' and insert 'plots omitted'

Page 18 Line 415 Delete 'so'

Page 18 line 417 Delete 'Yet,'

Page 19 Line 438 Delete 'from' and insert 'of', delete 'measurements' and insert 'measurements as well as'

Response: *We changed all four.*

References:

- Svenning, J.-C. & Sandel, B. *Disequilibrium vegetation dynamics under future climate change. American Journal of Botany* 100, 1266-1286, doi:10.3732/ajb.1200469 (2013).

REVIEWERS' COMMENTS:

Reviewer #1 (Remarks to the Author):

I have carefully read the revised version of the ms and the rebuttal letter. I'm satisfied with the responses that authors give to my former comments, thus, I can only congratulate authors for such excellent piece of work.

Reviewer #2 (Remarks to the Author):

I'm Reviewer 2. I'm pleased with the authors responses to my points and pleased with the revisions that have been made. I do believe the clarity of the paper has been improved and that this will be an important paper for the field.

Reviewer #3 (Remarks to the Author):

I'm satisfied that my comments have been dealt with adequately in the revised manuscript.

Trevor Booth

Response to referees

Reviewer #1 (Remarks to the Author):

I have carefully read the revised version of the ms and the rebuttal letter. I'm satisfied with the responses that authors give to my former comments, thus, I can only congratulate authors for such excellent piece of work.

Response: Thank you very much!

Reviewer #2 (Remarks to the Author):

I'm Reviewer 2. I'm pleased with the authors responses to my points and pleased with the revisions that have been made. I do believe the clarity of the paper has been improved and that this will be an important paper for the field.

Response: Thank you very much!

Reviewer #3 (Remarks to the Author):

I'm satisfied that my comments have been dealt with adequately in the revised manuscript.

Trevor Booth

Response: Thank you very much!